# Oligomerization state of the functional bacterial twin-arginine translocation (Tat) receptor complex

Ankith Sharma [1], Rajdeep Chowdhury[1] & Siegfried M. Musser [1]✉

The twin-arginine translocation (Tat) system transports folded proteins across bacterial and plastid energy transducing membranes. Ion leaks are generally considered to be mitigated by the creation and destruction of the translocation conduit in a cargo-dependent manner, a mechanism that enables tight sealing around a wide range of cargo shapes and sizes. In contrast to the variable stoichiometry of the active translocon, the oligomerization state of the receptor complex is considered more consistently stable but has proved stubbornly difficult to establish. Here, using a single molecule photobleaching analysis of individual inverted membrane vesicles, we demonstrate that Tat receptor complexes are tetrameric in native membranes with respect to both TatB and TatC. This establishes a maximal diameter for a resting state closed pore. A large percentage of Tat-deficient vesicles explains the typically low transport efficiencies observed. This individual reaction chamber approach will facilitate examination of the effects of stochastically distributed molecules.

[1] Department of Molecular and Cellular Medicine, Texas A&M University, School of Medicine, 1114 TAMU, College Station, TX 77843, USA.
✉email: smusser@tamu.edu

The Tat machinery has the unusual capacity to transport folded proteins across energetic membranes without collapsing ion gradients[1–6]. It accomplishes this without requiring any nucleoside triphosphates (NTPs) and with the unique requirement that a proton motive force (pmf) is essential for transporting all substrates[3,7,8]. Despite the seemingly complex physical problem of conveying a large object across a membrane while maintaining the permeability barrier, a minimal Tat system requires only two membrane proteins, TatA and TatC, e.g., as found in some gram-positive bacteria and archaea[9,10]. Nonetheless, the best-studied Tat systems are found in *Escherichia coli* and in plant thylakoids, both of which additionally contain TatB, a TatA-like membrane protein. Though *E. coli* also contains a fourth protein, TatE, which is homologous to TatA and TatB[6,11,12], *E. coli* TatABC comprise a common minimal system[6,7,12–17]. An early step in the transport cycle is recruitment of substrates with a twin-arginine signal peptide (containing an (S/T)RRXFLK consensus motif) by the Tat receptor complex[6,18]. However, transport cannot occur without a TatA-rich assembly, typically described as resulting from the recruitment of TatA to the receptor-substrate complex[19–21]. This active translocon is proposed to have a variable composition, allowing it to accommodate a range of mature domain sizes and shapes[22–24]. In contrast, the receptor complex is generally assumed to have a fixed composition, yet it has been surprisingly difficult to establish its oligomerization state[21,25–30].

TatC has six transmembrane helices that generate a glove-like groove within the membrane and forms the bulk of the signal peptide binding pocket[31,32]. Membranes containing TatB and TatC are sufficient for recruiting Tat substrates, establishing that these proteins form the core of the receptor complex[16,33,34], and these are generally agreed to be present in a 1:1 ratio[21,35,36]. Whether TatA is a normal constituent of the receptor complex has been controversial; at present, the weight of evidence leans toward some TatA in the receptor complex[16,26–28,30]. More recently, it was discovered that TatA, TatB, and TatE occupy very similar positions on TatC[37], suggesting that TatE is also part of the receptor complex, or that TatA takes the place of TatE when the latter is absent. Thus far, there is no evidence that any additional TatA and/or TatE within the receptor complex is in excess of one molecule per TatBC heterodimer. Thus, their presence would be insufficient to substantially increase the size of the receptor complex, although they could influence oligomerization state estimates. Due to the longstanding uncertainty regarding the presence of TatA and TatE in the receptor complex, its oligomerization state is typically discussed in terms of the number of TatBC heterodimers. Thus, the possible oligomerization states discussed throughout this paper (trimer, tetramer, etc.) refer to the oligomerization state of the TatBC heterodimer within the Tat receptor complex.

The receptor complex oligomerization state is fundamental to a mechanistic understanding of the transport mechanism. Considering that the receptor-bound signal peptide is inserted about halfway across the membrane[16], a fundamental structural issue is whether the signal peptide binding site is exposed to the membrane lipid interior, which it must be for a monomeric receptor complex, or whether it could be protected within the interior of a higher-order receptor oligomer[38]. For lipid-exposed binding site models, recruitment of TatA could, in principle, generate a translocation conduit with minimal (if any) conformational changes within the receptor complex. In contrast, for interior binding site models, substantial conformational changes appear necessary to rearrange to create a channel de novo, or to gate the channel open and closed, possibly also increasing the diameter of resting, but blocked, pore[28,38–40]. Notably, higher-order oligomers could form larger 'resting' pores, although they present a more substantial challenge to seal against ion leaks.

The receptor complex oligomerization state gleaned from various studies includes a dimer, trimer, tetramer, heptamer, and octamer, although trimer and tetramer models are favored more recently[21,25–30,35,36]. Large ~440–700 kDa complexes obtained upon detergent extraction[21,29,30,35,36,41] can overestimate oligomer size due to the lipid and detergent content of the analyzed micelles. Crosslinking approaches[26–28] can either under- or overestimate complex size due to insufficient crosslinks or crosslinks to unrelated proteins. Co-evolution analysis requires inferring the correct structural model and multiple oligomeric forms may be equally viable[28]. While conflicting conclusions over oligomerization state can certainly point toward variable compositions as conditions change, we considered that direct measurement by molecular counting in the native membrane environment would provide a more robust result.

Here, we optimized the overproduction level such that at most a few Tat receptor complexes are recovered per inverted membrane vesicle (IMV). Assuming that the cell resuspension and IMV formation processes do not disrupt the Tat receptor oligomer, the receptor complex oligomerization state in its native lipid environment can be inferred from the distribution of the Tat proteins in IMVs. Using a molecular counting approach, we show that in IMVs from cells overproducing TatABC, both TatB and TatC are present in four copies per receptor complex, the number of receptor complexes per IMV is ~1, and a large fraction of IMVs are devoid of Tat receptor complexes.

## Results

**Experimental design.** An inherent underlying assumption when purifying membrane proteins with detergents is that the extraction process does not influence the composition of the protein complex. Verification that this indeed so requires comparison with numerous controls. We avoided this complication by assaying the proteins present within native membrane vesicles. The formation of IMVs involves breaking the cell membrane into numerous pieces, which individually reseal to produce a membrane bilayer separating two aqueous compartments that in most cases is topologically inverted (the cytoplasmic leaflet is now on the outside)[7]. Thus, we expected that protein complexes present within the original cytoplasmic membrane would stochastically partition into the IMVs. Once the complexes were distributed in the IMVs, our counting assay does not rely on the protein assemblies remaining intact, merely that the individual polypeptides are expected to remain associated with the same IMV.

The overall goal of this study was to discern the number of TatB and TatC molecules present in the *E. coli* Tat receptor complex. We reasoned that by generating a low average number of receptor complexes per IMV via an appropriate overproduction level, the number of TatB or TatC molecules could be counted by single-molecule photobleaching methods on individually assayed IMVs. Our primary analytical assumption was that histogram distributions summarizing the number of observed steps in photobleaching traces were expected to be dependent on the oligomer size. For example, a tetramer can be expected to give rise to spots with a majority of fluorescent intensities corresponding to four, eight, twelve, etc. molecules. Thus, we expected to be able to determine for the first time the average number and distribution of TatB and TatC molecules and Tat receptor oligomers per IMV. To achieve this, we separately C-terminally tagged TatB and TatC with the fluorescent protein mNeonGreen (mNeon)[42] (Supplementary Fig. 1). IMVs were isolated from cells with Tat receptor complexes containing either TatB^mNeon or TatC^mNeon, the IMVs were incubated with the CellMask membrane stain, and the IMVs were then attached to microscope coverslips for photobleaching analysis (Fig. 1).

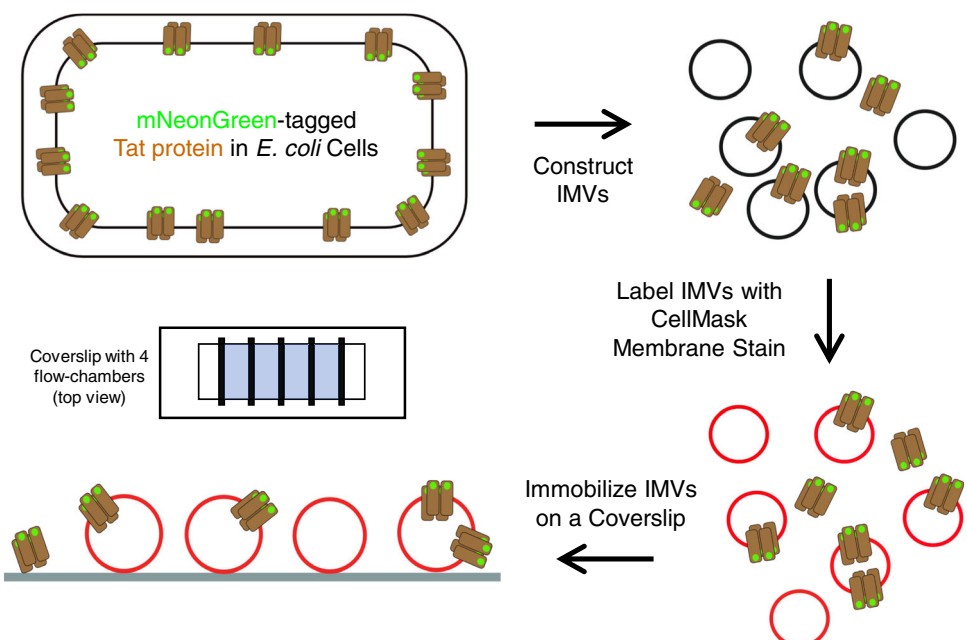

**Fig. 1 Experimental strategy.** IMVs were obtained from *E. coli* cells overproducing TatAB^mNeon^C or TatABC^mNeon^. After staining the membrane with CellMask, the IMVs were added to a flow chamber constructed on a microscope coverslip. The adsorbed IMVs were washed before analysis.

**Suitability of IMVs with mNeon-labeled Tat receptor complexes for molecular counting analysis.** To ensure complete labeling of the TatB and TatC proteins, the TatABC proteins were overproduced from a plasmid in the deletion strain MC4100ΔTatABCDE (a.k.a. DADE)[43]. IMVs produced from such cells are considered Tat++ IMVs[44]. A 1 h induction time yielded a low average number of TatB^mNeon^ or TatC^mNeon^ molecules per IMV, which was necessary for molecular counting. This is a substantially lower production level than for IMVs used for typical in vitro transport assays (1 h vs. 4 h overproduction time)[7,17,45,46]. Not surprisingly, then, these IMVs containing TatB^mNeon^ or TatC^mNeon^ exhibited reduced transport efficiencies compared with those obtained after 4 h induction (Fig. 2a, b). No noticeable cleavage of the TatB^mNeon^ and TatC^mNeon^ proteins in IMV preparations was observed (Supplementary Fig. 2), indicating their suitability for molecular counting. Since the transport efficiency was similar using: (1) TatB^mNeon^ or wildtype TatB (Fig. 2a, b); (2) TatB^mNeon^ or TatC^mNeon^ (Fig. 2a); and (3) TatB^mCherry^ or TatC^mCherry^ compared with wildtype TatB or TatC[17], we consider it reasonable to assume that the oligomerization state of the Tat receptor complex was not perturbed by the mNeon tags. The average diameters of TatB^mNeon^ and TatC^mNeon^ IMVs were 105 nm and 100 nm, respectively, as determined by dynamic light scattering (DLS; Fig. 2c). This small size indicates that individual IMVs should be visualized essentially as diffraction-limited spots when using light microscopy. Further, all fluorophores present in such IMVs are expected to be well-focused and simultaneously detectable in a single fluorescent spot. Individual complexes or molecular diffusion were not resolvable in our experiments. Instead, our approach relies on the assumption that oligomers held together by strong interactions will partition into IMVs as intact assemblies, and that different oligomerization models will yield experimentally distinguishable photobleaching characteristics.

**Colocalization of TatB^mNeon^ or TatC^mNeon^ with membrane stains.** To exclude Tat aggregates/assemblies from our analysis that were not associated with IMVs, we reasoned that the IMVs could be identified with the hydrophobic membrane stain Cell-Mask, and only spots that appeared in both channels would be analyzed. CellMask stained IMVs were adsorbed to plasma-treated coverslips at a dilution suitable for obtaining a sparse distribution of well-isolated fluorescence spots, which were visualized by narrow-field epifluorescence microscopy[47]. The mNeon and CellMask channels allowed independent identification of spots containing the labeled Tat protein and CellMask-positive entities, respectively (Fig. 3a). Initial colocalization analysis revealed that: (i) ~40% of CellMask-positive entities colocalized with mNeon-tagged Tat protein and (ii) 30–50% of the mNeon-tagged Tat protein was associated with a CellMask-positive entity (Fig. 3b). We hypothesized that the mNeon-tagged protein that did not colocalize with a CellMask-positive entity reflected soluble Tat protein present within the cytoplasm that had not yet been incorporated into the membrane before cell rupture, and that had copurified with the IMVs. To test this, bacterial cells were allowed to grow for another 2 h without the arabinose inducer ("2 h chase"), thus providing additional time for membrane incorporation of the produced mNeon-tagged Tat proteins. Under these conditions, the percentage of mNeon-tagged Tat proteins that colocalized with a CellMask-positive entity increased to ~50–80% (Fig. 3b). These data support the hypothesis that mNeon-tagged Tat protein that does not colocalize with the CellMask stain had not yet undergone membrane integration. The colocalization of the CellMask-positive entities with mNeon decreased to ~20–30% (Fig. 3b), consistent with a dilution of the Tat receptor complexes in the cytoplasmic membrane due to additional cell divisions (Supplementary Fig. 6).

We also compared the staining patterns of two other hydrophobic membrane dyes, FluoVolt and BODIPY FL C16, with that of CellMask. In principle, all three membrane stains should partition into all IMVs, thus yielding equivalent staining patterns. Using wildtype TatABC IMVs, ~58% of FluoVolt spots and ~38% of BODIPY FL C16 spots colocalized with CellMask spots; conversely, ~56% and ~47% of CellMask-positive spots colocalized with FluoVolt and BODIPY FL C16 spots, respectively (Fig. 3c). These are strikingly different from the expected 100% colocalization, assuming that all three dyes stain lipid membranes

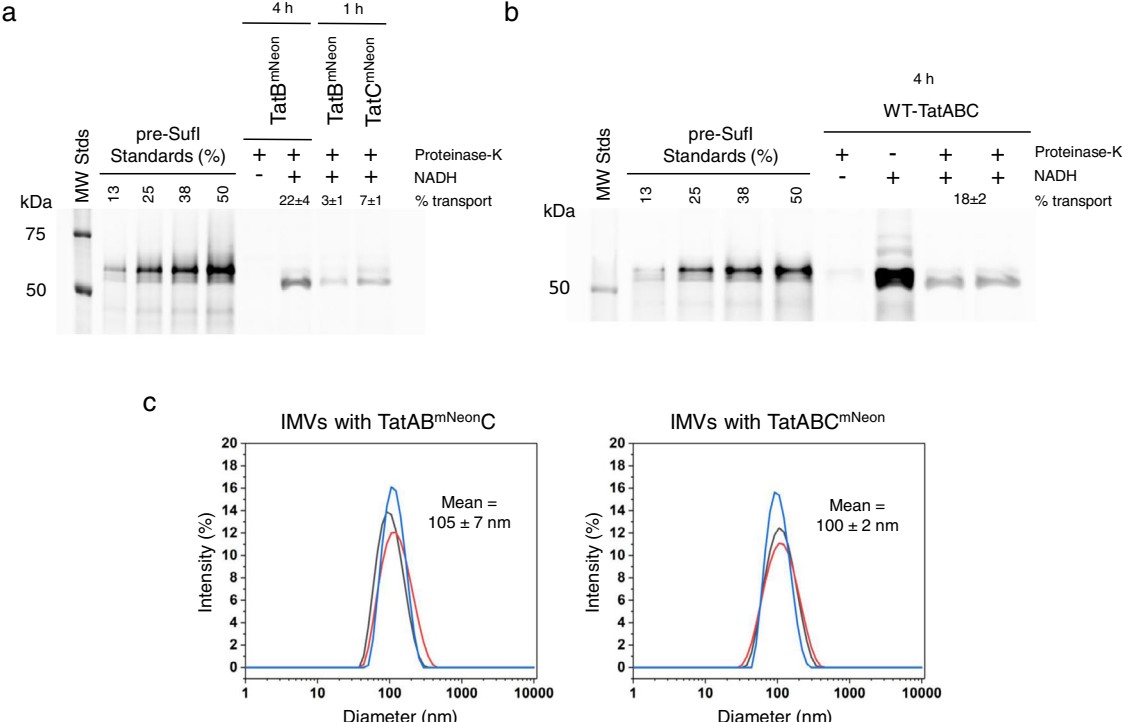

**Fig. 2 Transport efficiency and size of IMVs with TatAB^mNeon^C or TatABC^mNeon^. a, b** In vitro transport assays. The amount of SufI(IAC)^Alexa647^ (50 nM) transported into Tat^++^ IMVs was determined after 20 min at 37 °C. Transport efficiency (% transport) was estimated from the amount of protease-protected mature-length protein compared with the amount of total added precursor protein (pre-SufI standards). The necessary pmf was generated by the addition of NADH (4 mM). A 4 h induction period yielded a similar transport efficiency for TatAB^mNeon^C (**a**) and wildtype TatABC (Tat^++^) IMVs (**b**). SDS-PAGE gels were analyzed by direct in-gel fluorescence imaging ($\lambda_{ex}$ = 488 nm). A substantially lower transport efficiency was observed for a short (1 h) overproduction time (**a**), which was required to ensure a low number of Tat receptor complexes per IMV. All transport efficiencies were calculated from duplicate assays. **c** IMV size. Diameter distributions of three independent preparations of IMVs containing either TatAB^mNeon^C or TatABC^mNeon^, as determined by DLS. A consistent average diameter of ~100 nm was obtained.

and only lipid membranes. These data establish that the three hydrophobic dyes examined have distinct staining patterns, likely staining other hydrophobic copurifying entities in addition to the IMV membrane. Since the FluoVolt or BODIPY FL C16 dyes cannot be used with mNeon as all three fluorophores are excited at 488 nm, we used CellMask to eliminate non-colocalizing TatB^mNeon^ or TatC^mNeon^ fluorescent spots, which were generally of low intensities and therefore consistent with largely non-integrated monomeric proteins. This analysis also suggests that CellMask potentially overestimates the number of IMVs present in a microscope field.

**Step analysis of photobleaching traces**. To establish an appropriate step-counting protocol for photobleaching analysis of the labeled Tat proteins, photobleaching traces for mNeon monomers, dimers, and tetramers were obtained (Fig. 3d and Fig. 4a). While fluorescence blinking was observed, the combination of intensity and observable steps generated confidence in a manual approach to step counting. For the mNeon-labeled Tat proteins that colocalized with CellMask, the majority of mNeon fluorescence spots were approximately diffraction-limited with variable initial intensities. Characteristic stepwise patterns of photobleaching with a variable number of steps were observed (Fig. 4b). Some larger, intensely fluorescent spots were also observed (most of which were probably aggregates). For ≤~8 steps, the manual step-counting approach used for the mNeon standards was generally suitable. However, we also used a pairwise difference distribution (PDD) function[48] for a more quantitative and less biased step-counting analysis (Fig. 5).

The step distributions for IMVs containing TatB^mNeon^ or TatC^mNeon^ reveal that all possibilities from 1 to 8 steps were observed (Fig. 6). While >8 steps were occasionally observed, eight steps were deemed the limit where manual and PDD step-picking was reliable, so anything above this was not included in our analysis. Assuming the simplest model, i.e., that the receptor complex consists of only one TatBC heterodimer, the step histograms were fit assuming that these complexes were Poisson distributed into the IMVs. Since control experiments revealed that ~2.7% of CellMask-positive entities using wildtype TatABC IMVs had detectable fluorescence in the mNeon channel, which almost always bleached in a single step (Supplementary Fig. 4), the 1 step data bin was not included in the fitting routine. While these fits for TatB^mNeon^ or TatC^mNeon^ photobleaching histograms predict an average number of monomeric receptor complexes per IMV of ~3 (Fig. 6), they also predict a low percentage of empty vesicles (~4%), which is inconsistent with the high percentage of CellMask-positive spots that were devoid of fluorescent Tat proteins (>~70%; Fig. 3b). In addition, photobleaching histograms for dimeric and tetrameric mNeon exhibited a substantial number of molecules exhibiting steps below the nominal oligomer size (Fig. 7a–c), which could not be explained by fragments in the protein preparations (Supplementary Fig. 8). Thus, a correction involving a photophysical explanation seemed more likely, which is described in the sections that follow.

**Fluorescence detection efficiency (FDE) of mNeon**. Fluorescent proteins are not always detectable, a phenomenon that is typically

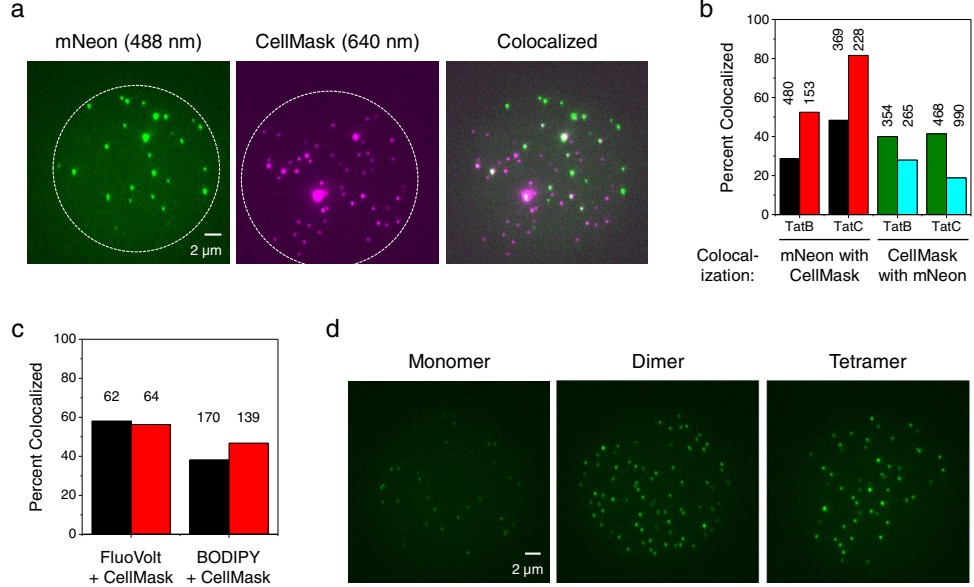

**Fig. 3 Single particle imaging of IMVs with TatABC^mNeon and mNeon standards. a** Adherent IMVs. IMVs containing TatABC^mNeon stained with the CellMask lipid dye were visualized on coverslips using 488 nm (mNeon, green channel) and 640 nm (CellMask, red channel) excitation. The mNeon was imaged first and photobleached before imaging with the red channel. A very low amount of the free CellMask dye sticks to the coverslip surface (Supplementary Fig. 3), indicating that CellMask reports on the presence of material in the IVM preparation. IMVs with wildtype TatABC exhibit almost no fluorescence in the green channel (Supplementary Fig. 4). Only colocalized spots (TatC^mNeon within a CellMask-positive entity) were used for photobleaching analysis. The IMVs imaged here were obtained from cells in which TatABC^mNeon was overproduced for 1 h with 0.7% arabinose. IMVs prepared in this manner yielded an ~50% colocalization of mNeon with CellMask. This colocalization percentage is not increased with a 10-fold higher concentration of CellMask (Supplementary Fig. 5). The dashed white circles indicates the approximate illumination area using narrow-field epifluorescence[47]. By including a "2 h chase" during cell growth, the number of mNeon spots per field decreased but the colocalization percentage with CellMask increased (Supplementary Fig. 6). **b** Colocalization percentages of labeled Tat proteins and CellMask. IMV preparations containing TatAB^mNeonC or TatABC^mNeon were incubated with CellMask, and then visualized in the green and red channels. Quantified are: (i) spots visualized in the mNeon channel that were CellMask-positive using IMVs from 1 h (black) or 1 h + "2 h chase" (red) TatABC overproduction times; and (ii) spots visualized in the CellMask channel that were mNeon-positive using IMVs from 1 h (green) or 1 h + '2 h chase' (blue) TatABC overproduction times. Numbers indicate the total number of spots identified with the primary fluorophore. **c** Colocalization percentages of the FluoVolt and BODIPL-FL C16 dyes with CellMask. IMVs generated from wildtype TatABC overproduced for 1 h were co-incubated with two hydrophobic fluorescent stains, as indicated (see Supplementary Fig. 7). Data were quantified as the FluoVolt- or BODIPY FL C16-positive spots colocalizing with CellMask (black) and the CellMask-positive spots colocalizing with FluoVolt or BODIPY FL C16 (red). Numbers indicate the total number of spots identified with the primary fluorophore. CellMask (0.1×) and FluoVolt (2×) were used at the indicated dilutions of manufacturer stocks; BODIPY FL C16 was used at a final concentration of 100 pM from a DMSO stock. **d** Coverslip-adsorbed mNeon standards. Shown are monomer (Cys-free mNeon), dimer (disulfide dimer of mNeonC), and tetramer (disulfide dimer of 2xmNeonC) forms of mNeon, as described in Supplementary Fig. 8.

assumed to largely arise from deficiencies in fluorophore maturation[49–53]. A non-unity FDE results in undercounting the number of fluorescent protein domains present. For an FDE substantially less than 1, the number of steps corresponding to the nominal oligomer size may not be the most frequent observation (e.g., Fig. 7a), thereby complicating the identification of an unknown oligomeric state. Assuming that non-fluorescent mNeon domains were binomially distributed into mNeon tetramers, the FDE of mNeon in this oligomer was estimated as 0.76 (Fig. 7a, d). The FDE of mNeon in disulfide dimers of mNeon was ~0.82 (Fig. 7b, d), indicating a higher detection efficiency. In contrast, the FDE of mNeon in genetic dimers of mNeon was ~0.77 (Fig. 7c, d), consistent with the FDE for the mNeon tetramer, which was formed as a disulfide dimer of genetic mNeon dimers (Supplementary Fig. 8). These data suggest that the FDE is determined by the identity of the protein when overproduced in bacterial cells, rather than the final oligomeric complex analyzed. This conclusion suggested that the FDE determined for Tat fusions would differ from those determined for these mNeon standards.

**The Tat receptor complex is a tetramer with respect to both TatB and TatC.** The analysis of the IMV photobleaching

histograms was more complex than that for the mNeon oligomers since the IMVs contain variable numbers of Tat receptor complexes. We assumed a model in which the number of Tat receptor complexes was Poisson distributed into IMVs, and the number of active fluorophores in each oligomer was binomially distributed, as determined by the FDE. Thus, there were two fitting parameters (Poisson mean and FDE). We emphasize that the Poisson mean refers to the mean of the distribution of the number of Tat receptor complexes within the IMVs, not the mean number of photobleaching steps. For both TatB^mNeon and TatC^mNeon, the best fits resulted when four copies of these proteins were in each receptor complex (Fig. 8, Supplementary Fig. 10, and Supplementary Table 1). This conclusion was confirmed with duplicate data sets for each labeled Tat protein (Supplementary Fig. 11 and Supplementary Table 1), and for a PDD analysis of one TatB and TatC dataset both with and without a "2 h chase" (Fig. 8, Supplementary Fig. 12, and Supplementary Table 1).

As indicated earlier, the FDE for the mNeon standards varied depending on whether the mNeon protein was monomeric or dimeric when overproduced in E. coli. Not surprising, then, the FDE for mNeon was somewhat lower when it was fused to the Tat proteins (~0.6–0.7, assuming tetramers), indicating both that

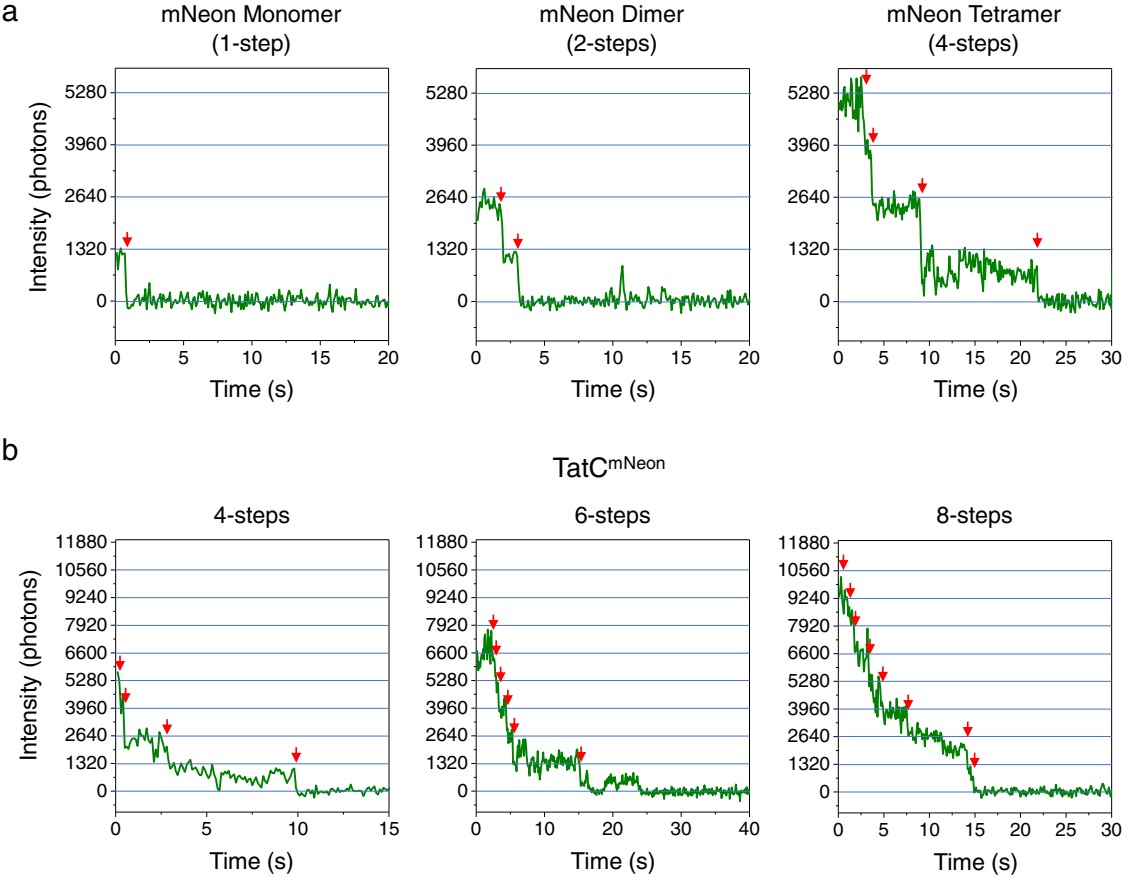

**Fig. 4 Manual identification of mNeon photobleaching steps. a** Sample photobleaching traces of mNeon standards, as identified in Fig. 3d. **b** Sample photobleaching traces of IMVs containing TatABC<sup>mNeon</sup>. For both **a**, **b**, steps are established at the times identified by the red arrows (manually identified). While blinking events occurred and caused deviations from regular steps, the combination of step-size and total intensity allowed the assignment of the total number of steps in the photobleaching traces. Individual traces were background subtracted and normalized to zero. Supplementary Fig. 9a demonstrates the linearity of photon counts with increasing oligomer size and that the average step was ~1320 photons (blue lines).

the attached Tat proteins affected fluorophore maturation, and that the FDE must be directly estimated from the experiment rather than from simple control proteins. We expect that both total molecular weight, as well as polypeptide composition, influence the fluorophore maturation of fluorescent proteins (Supplementary Fig. 13).

## Discussion

The reported work probed the oligomerization state of the Tat receptor complex in a native membrane environment by counting photobleaching steps. While conceptually straightforward, this approach typically encounters significant challenges: (i) the photobleaching steps must be distinguishable; (ii) the interpretation relies on the FDE (which can vary depending on the fusion protein); (iii) the number of expected steps is dependent on the number of complexes in an analyzed spot; and (iv) potential interference from background components must be accounted for. Point (i) was addressed by using mNeon, a bright, relatively photostable fluorescent protein[42]. While blinking was observed, this did not eliminate our ability to manually count steps, which was verified by similar results obtained through PDD analysis. Points (ii) and (iii) were addressed by including the FDE and complex distribution (Poisson mean) in the data analysis routine as fitting parameters. The FDE was dependent on the fusion construct, indicating differences in chromophore maturation efficiency for distinct fusion proteins. Background components (point iv) were eliminated by spot selection (e.g.,

colocalized "membrane" dye and mNeon fluorescence) or subtraction (e.g., single-step photobleaching spots on the coverslips or arising from non-tagged IMVs). Our analysis revealed that the Tat receptor complex including four copies of both TatB and TatC is the most parsimonious interpretation of the photobleaching data. We first expand on the support for this conclusion and then discuss the implications.

Our conclusion that the Tat receptor complex is tetrameric with respect to both TatB and TatC is primarily based on the finding that the root-mean-square deviation (RMSD) between the experimental data and the best fit for all data sets is lowest for this oligomerization model. Six distinct preparations of IMVs—three containing TatAB<sup>mNeon</sup>C and three containing TatABC<sup>mNeon</sup>—yielded the same conclusion (Supplementary Table 1). While there is a clear and significant break between the RMSD for tetramers and lower order oligomers (trimers and lower), the average RMSD for somewhat higher-order oligomers was similar to the tetramer fit, although it progressively increased (Fig. 8c). However, there are additional considerations. Maturation efficiencies (including for mNeon) are typically above 0.5[49–51,54–58], suggesting that dimer, trimer, and hexamer or higher models are unlikely (Fig. 8d). A functional analysis of TatC fusion proteins suggested that the TatC oligomerization state is an even number[59], thus arguing against (but not completely eliminating) the pentamer model (as well as other odd oligomerization states).

The high number of Tat-deficient IMVs also argues against the trimer and lower models. From the colocalization analysis, >50%

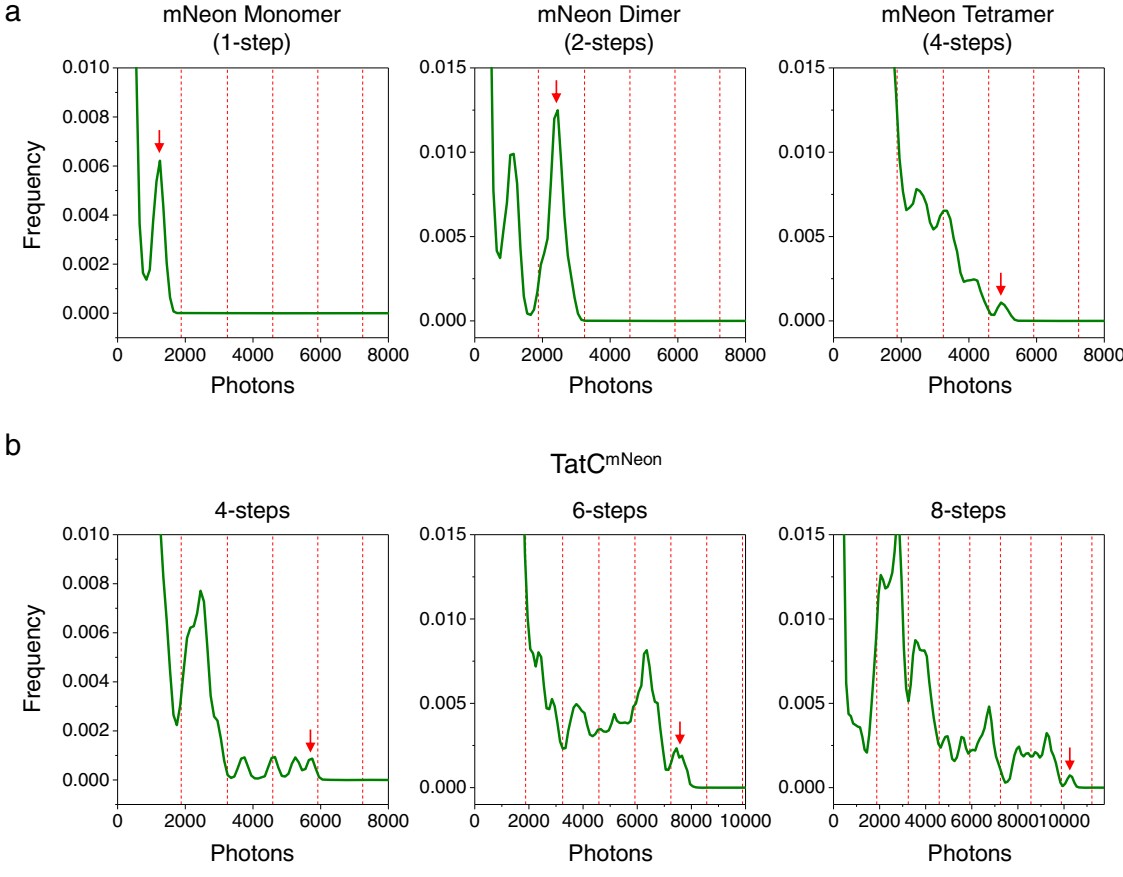

**Fig. 5 PDD identification of mNeon photobleaching steps. a**, **b** PDD histograms for the traces shown in Fig. 4. The last peak (red arrow) was used to determine the number of photobleaching steps in the trace. The boundaries between step assignments (red dashed vertical lines) are described in the Methods and Supplementary Fig. 9b.

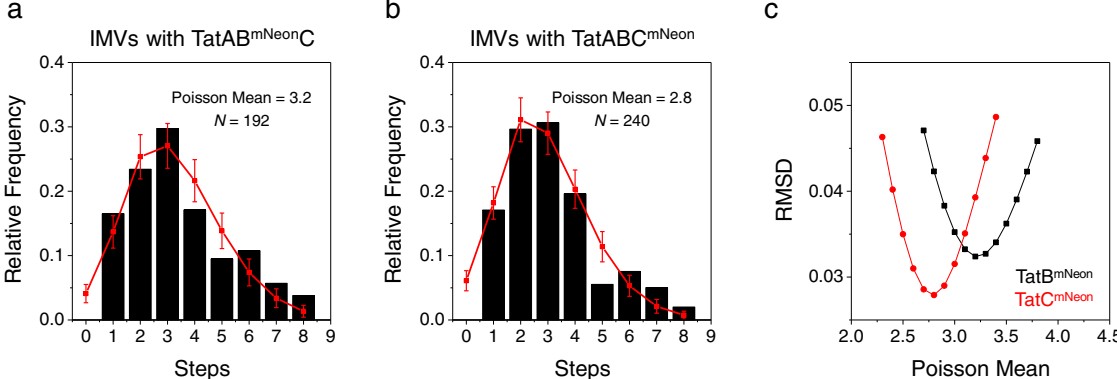

**Fig. 6 Photobleaching step histograms for IMVs with TatAB^mNeon^C or TatABC^mNeon^. a**, **b** Histograms of the number of steps observed in photobleaching profiles of IMVs containing TatAB^mNeon^C or TatABC^mNeon^. Tat proteins were overproduced for 1 h + '2 h chase', and the number of photobleaching steps were determined with the PDD function. The data (black) were fit assuming the simplest model, i.e., that the FDE = 1, that receptor complexes contain a single TatBC heterodimer, and that these are distributed into IMVs according to a Poisson distribution. The single-step bin contained contaminants that bleached in a single step, and occurred at a density of ~2 spots per microscope field (~2.7% of total CellMask-positive spots; see text and Supplementary Fig. 4). Consequently, the data were fit to step bins 2–8. The best fit (red) was determined by the least root-mean-square deviation (RMSD) between the experimental and expected values (Poisson probabilities). Error bars are standard deviations (SDs) obtained from 5000 simulated distributions of $N$ values each, where $N$ is the number of experimental measurements from a single IMV preparation, as indicated. **c** The optimal Poisson mean. The RMSD (step bins 2–8) for different Poisson mean values were calculated for the data in **a**, **b**, assuming that all mNeon proteins were actively fluorescent (FDE = 1). The minimum RMSD occurs at Poisson means of ~3.2 and ~2.8 for TatB^mNeon^ and TatC^mNeon^, respectively; the corresponding predicted distributions are shown in **a**, **b**.

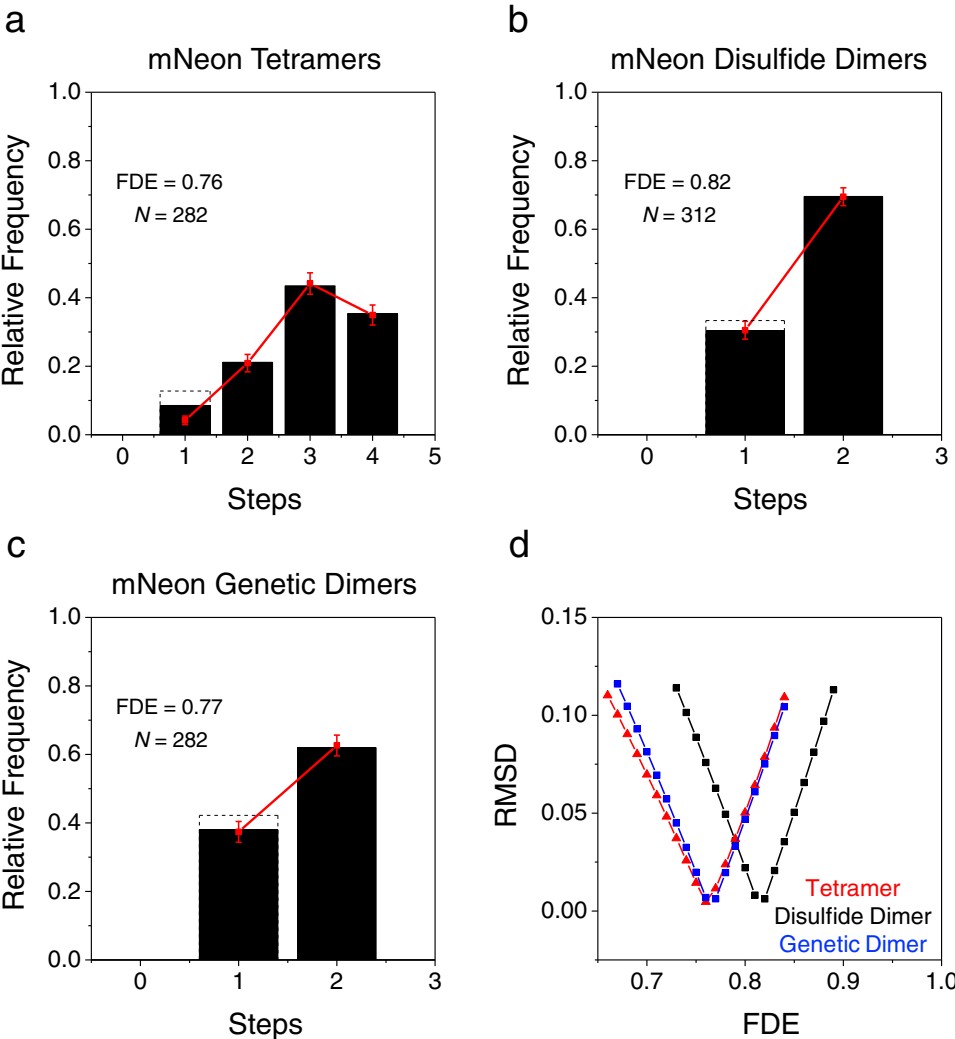

**Fig. 7 Photobleaching step histograms for mNeon tetramers and dimers. a** Photobleaching step histogram for mNeon tetramers (disulfide dimers of 2xmNeonC). The data (black) were fit assuming that the fluorescently active mNeon proteins were binomially distributed within the individual tetramers, yielding a best fit (red) fluorescence detection efficiency (FDE) of 0.76. The single step data were ignored for the fit shown, acknowledging the known presence of surface contaminants. The corrected single-step data (solid; original data shown as dashed box) obtained by subtracting the estimated number of contaminants (~1.9 spots/field; see Supplementary Fig. 3) agree better with the prediction from the fit. **b** Photobleaching step histogram for disulfide dimers of mNeonC. Assuming a binomial distribution of the fluorescently active mNeon proteins, the corrected data (solid black) predict an FDE ≈ 0.82. The correction here (original data shown as dashed box) is substantially less than in **a** since the dimer single step data were a larger fraction of the total and the dimer spots were somewhat more densely distributed on the surface than the tetramer spots. **c** Photobleaching step histogram for mNeon genetic dimers. The FDE ≈ 0.77 obtained here is lower than that obtained for the disulfide dimers (**b**) but similar to that of the tetramers (**a**), which were generated from genetic dimers. **a–c** Error bars are standard deviations (SDs) obtained from 5000 simulated distributions of $N$ values each, where $N$ is the number of experimental measurements from one protein preparation, as indicated. **d** The optimal FDE. The RMSDs for different FDE values were calculated for the data in **a–c**, as indicated. Tetramer fits were to step bins 2–4; dimer fits were to the corrected data.

of the CellMask-positive structures did not contain an mNeon-labeled Tat protein (Fig. 3b). This is substantially higher than the number of empty vesicles predicted for the trimer and lower oligomerization models (in most cases, <20%; Supplementary Table 1). The low colocalization of Tat proteins with CellMask (Fig. 3d) is not a consequence of a low dye concentration, as a 10-fold increase in the concentration of CellMask yielded the same colocalization percentage (Supplementary Fig. 5). The assumption that CellMask identifies only lipid-containing structures were challenged by the finding that two other hydrophobic dyes that were presumed to also partition into membranes yielded distinct staining patterns (Fig. 3c; Supplementary Fig. 7). These data argue that the number of CellMask-positive spots overestimates the number of IMVs present. Consequently, we conclude that the

~40–50% empty vesicles predicted from the tetramer and higher oligomer models are reasonable for our IMV preparations. An additional possible contributing factor for the low TatB/C and CellMask colocalization percentage is that the Tat receptor complexes were not homogeneously distributed within the originating membrane[23,60]. IMVs were constructed from spheroplasts, however, so it is the distribution within spheroplasts, not within intact cells, that dictates the partitioning into the resultant IMVs. We found that both TatB and TatC are fairly homogeneously distributed within spheroplast membranes (Supplementary Fig. 14). This homogeneity is a key assumption for the model that the Tat receptor complexes were Poisson distributed into the IMVs.

The low but measurable transport efficiencies (≤~7%) of the IMVs used for photobleaching studies (Fig. 2a) suggest that not

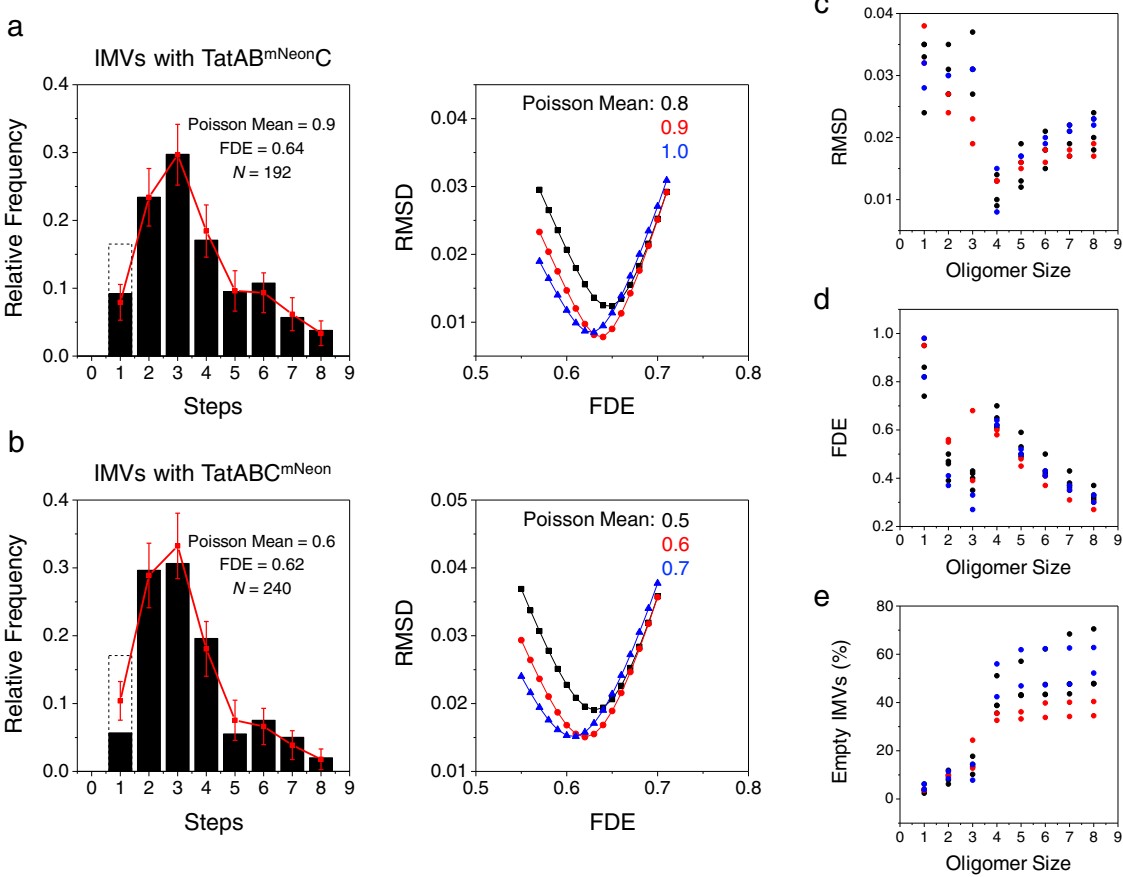

**Fig. 8 The tetrameric Tat receptor complex. a, b** Best fits to photobleaching data for IMVs containing TatAB$^{mNeon}$C or TatABC$^{mNeon}$. These are the same data as shown in Fig. 6a, b. By varying the assumed oligomerization state (monomers to octamers), the Poisson mean (which describes the expected distribution pattern for the number of Tat complexes in the IMVs), and the FDE, the best fits to the experimental photobleaching data for IMVs containing TatAB$^{mNeon}$C (**a**) or TatABC$^{mNeon}$ (**b**) indicate that the Tat receptor complex includes four copies each of TatB and TatC. Sample fits for other oligomerization states are shown in Supplementary Fig. 10; parameters for the different oligomeric fits are reported in Supplementary Table 1 (data in blue). The single-step data in the histograms are corrected (solid black, see Fig. 6); uncorrected values are also indicated (dashed black boxes). Only step bins 2–8 were included in the RMSD for the best fits shown (red); inclusion of the corrected single step bin for both data sets leads to identical conclusions (see Supplementary Table 1). Error bars are standard deviations (SDs) obtained from 5000 simulated distributions of $N$ values each, where $N$ is the number of experimental measurements from a single IMV preparation, as indicated. **c–e** Summary of fit parameters and predicted empty IMVs. The RMSDs (**c**), FDEs (**d**), and predicted percentage of empty IMVs (**e**) assuming various oligomerization states for three preparations each of IMVs containing TatAB$^{mNeon}$C or TatABC$^{mNeon}$ show a transition between the trimer and tetramer models. The best fit (lowest residuals between the fit and the data using RMSD) was obtained assuming a tetramer [as shown in **a**, **b**]. Raw data are summarized in Supplementary Table 1 (with identical color coding), which describes the different IMV preparations and how the data were analyzed.

all of the TatB/C spots that colocalized with CellMask (≥~30%) represent transport competent IMVs. In addition to a Tat receptor complex, transport competence requires sufficient TatA and respiratory proteins (to generate the necessary pmf) to partition into the IMV. Since higher TatABC expression levels increase transport competence, TatA is likely the primary limiting protein, although multiple receptor complexes per IMV could also increase transport efficiency. Considering a cell size of ~2 (length) × 1 (width) μm for MC4100, ~200 IMVs (100 nm diameter) could be obtained from a single *E. coli* bacterium. A Poisson mean of 1 for the number of complexes per IMV (Supplementary Table 1) therefore implies ~200 Tat receptor complexes per *E. coli* cell. Since overproduction of TatABC is necessary to observe in vitro transport with IMVs[7], this value of 200 is an upper limit for the number of wildtype Tat receptor complexes in non-overproducing cells, likely by a wide margin[19]. In short, the large number of IMVs obtained per *E. coli* cell and the low wildtype levels of Tat proteins indicates that significant Tat protein overproduction levels are required to obtain high overall transport efficiencies as well as transport into most IMVs.

The mechanism of Tat translocation remains under debate. Nonetheless, most recent models place the signal peptide binding site on the inside of the Tat receptor complex[26–28,61,62]. In such a mechanism, cargo translocation would either occur through the center of the receptor complex, or subsequent to a substantial conformational change wherein the necessary conduit is generated in a more external location. In the former case, expansion of the resting pore would be necessary, likely upon recruitment of TatA, as a tetrameric receptor complex is not expected to be able to accommodate all folded protein cargos. In either case, it is unclear if a channel exists in the center of the tetrameric receptor complex, which, if it exists, would necessarily need to be gated closed to prevent collapse of the pmf. Establishing that the Tat receptor complex contains four copies of both TatB and TatC

establishes a critical framework for addressing these structural and functional issues.

## Methods

**Bacterial strains, growth conditions, and plasmids.** The *E. coli* strains MC4100ΔTatABCDE (DADE), JM109, and BL21(λDE3) were described earlier[63–65]. Since MC4100 is an *araD* mutant strain whose growth is known to be inhibited by L-arabinose, an arabinose-resistant DADE strain was generated as described earlier[44] and used for Tat overproduction. JM109 was used for clone amplification and plasmid maintenance. BL21(λDE3) cultures producing pre-SufI and the mNeon fluorescent standards from the indicated plasmids were grown in Luria-Bertani (LB) medium[66] at 37 °C. Tat proteins were expressed in the DADE strain and grown in a modified low-salt LB media with glycerol (1% bactotryptone, 0.5% yeast extract, 0.25% NaCl and 0.5% glycerol). All cultures were supplemented with the appropriate antibiotic (30 µg/mL kanamycin or 50 µg/mL ampicillin). All new plasmids were submitted to Addgene, and their construction is described in the history of the linked SnapGene files. Coding sequences were verified by DNA sequencing. Plasmid constructions are briefly outlined below, and the encoded amino acid sequences are indicated in Supplementary Fig. 1.

The pET-28a_mScarlet plasmid was created by amplifying mScarlet from pCytERM mScarlet N1 (Addgene #85066) with an N-terminal 6xHis-tag and an SpeI site between the 6xHis-tag and mScarlet and inserting the H6-SpeI-mScarlet PCR product into pET28a (Novagen #69864) using NcoI and BamHI. The mNeon coding sequence amplified from plasmid L40C-CRISPR.EFS.mNeon (Addgene #69146) was then used to replace mScarlet using SpeI and BamHI, generating plasmid pH6-mNeon.

The mNeon coding sequence was amplified from pH6-mNeon using primers that introduced a C-terminal cysteine residue, and the PCR product was then re-inserted back into the same plasmid using SpeI and BamHI, generating plasmid pH6-mNeonC.

The mNeon coding sequence was amplified from the pH6-mNeon plasmid with an SpeI site at both ends, and the PCR product was inserted upstream of the existing mNeon gene in the pH6-mNeon plasmid using SpeI, generating plasmid pH6-2xmNeon (correctly oriented product).

The plasmid pH6-2xmNeonC was created identically as plasmid pH6-2xmNeon except that the PCR product containing the second mNeon gene was inserted upstream of the existing mNeon copy in the pH6-mNeonC plasmid using SpeI.

The mNeon coding sequence was amplified from the pH6-mNeon plasmid, and the PCR product was used to replace mCherry in pBAD_TatAB^mCherryC[17] using KpnI and SpeI, generating plasmid pTatAB^mNeonC.

The pTatABC plasmid[67] was modified to contain restriction sites after each Tat gene using the QuikChange protocol. The new plasmid (pTatA_kB_xC_s) contained a KpnI restriction site after TatA, an XbaI site after TatB, and a SalI site after TatC. The mNeon coding sequence was amplified from the pH6-mNeon plasmid with SalI and SphI sites at the two ends, and the PCR product was inserted downstream of TatC in pTatA_kB_xC_s using SalI and SphI, generating plasmid pTatABC^mNeon (correctly oriented product).

**Protein production and purification.** Monomeric and dimeric mNeon were obtained from 500 mL cultures inoculated with 5 mL starter cultures grown for 16 h. Protein production was induced at $OD_{600} \approx 0.6$ with 1 mM isopropyl β-D-1-thiogalactopyranoside (IPTG) and growth was continued for 3 h at 37 °C. Cell pellets were stored at −80 °C until purification. After frozen cells were thawed on ice, they were rapidly resuspended by adding 40 mL lysis buffer (100 mM Tris-HCl, 250 mM NaCl, 1% Triton X-100, pH 7.5) containing 1× CelLytic B (Cat. #C8740, Sigma), protease inhibitors [10 mM phenylmethylsulfonyl fluoride (PMSF), 20 µg/mL trypsin inhibitor, 10 µg/mL leupeptin and 10 µg/mL pepstatin], 20 µg/mL DNase, and 10 µg/mL RNase. The cell suspension was cleared by centrifugation at $40,000 \times g$ for 20 min at 4 °C. The supernatant was loaded onto a $10 \times 1$ cm column with 2 mL Ni-NTA Superflow resin (Cat. #30230, Qiagen) that was pre-equilibrated with Buffer A (10 mM Tris-HCl, 1 M NaCl, 0.1% Triton X-100, pH 8.0). The resin was washed with 30 mL of Buffer A, 30 mL of 10 mM Tris-HCl, 1 M NaCl, pH 8, 10 mL of 10 mM Tris, 0.1 M NaCl, pH 8, and 5 mL of 10 mM Tris-HCl, 50 mM NaCl, 50% glycerol, pH 8. The bound protein was eluted with 500 mM imidazole, 50 mM NaCl, 50% glycerol, pH 8. Eluates were analyzed by SDS-PAGE to check for purity and concentration. For disulfide dimers of mNeonC and 2xmNeonC (yielding disulfide dimers and tetramers, respectively), the mNeon-containing elution fractions were incubated overnight at 4 °C and then purified using size exclusion chromatography with FPLC buffer (50 mM Tris-HCl, 150 mM NaCl, pH 7.5) the next day (Amersham Pharmacia Biotech AKTAdesign system with Superdex 75 column for the disulfide dimer, and Bio-Rad NGC system with Enrich SEC 650 column for the tetramer). The mNeon genetic dimer (2xmNeon) was also purified via size exclusion chromatography using FPLC buffer (Bio-Rad NGC system with Enrich SEC 70 column).

The Tat substrate pre-SufI(IAC) was overproduced from plasmid p-pre-SufI(IAC) (Addgene #168516). A starter culture (5 mL) grown for 16 h was transferred to a 500 mL culture. Protein production was induced at $OD_{600} \approx 0.6$ with 1 mM IPTG, and 25 mL 0.5 M CAPS, pH 10 was added to raise the culture pH[16]. After growth for 2 h at 37 °C, cells were recovered ($5000 \times g$, 10 min, 4 °C) and then stored at -80 °C until purification. The pre-SufI(IAC) protein was purified via Ni-NTA chromatography following the same procedure used for the mNeon protein standards. To label with a fluorescent dye, purified pre-SufI(IAC) was first incubated with 1 mM *tris*[2-carboxyethylphosphine] hydrochloride for 10 min, and then with a 50-fold molar excess of Alexa647 maleimide at room temperature (RT) in the dark. After 15 min, the reaction was quenched with 10 mM β-mercaptoethanol (βME). The mixture was diluted 10-fold to reduce the imidazole concentration and then re-purified by Ni-NTA chromatography.

**Isolation of IMVs.** IMVs were prepared from the DADE strain transformed with pTatAB^mNeonC or pTatABC^mNeon. A starter culture (5 mL) grown overnight at 37 °C was used to inoculate 3 L ($6 \times 500$ mL) of modified low-salt LB media (described earlier). Tat protein production was induced at $OD_{600} \approx 0.6$ with 0.7% arabinose for 1 h or 4 h at 37 °C. In some cases, a 2 h "chase" was performed by spinning down the cells after 1 h of Tat protein production, resuspending the cells in fresh media, and incubating at 37 °C for 2 h. Cells were recovered ($5000 \times g$, 10 min, 4 °C) and then stored at −80 °C until the next day. IMVs were isolated as described[17], except that: (i) lysozyme was used at 1 mg/mL when making spheroplasts; (ii) the EDTA/lysozyme incubation time to make spheroplasts was 30 min; (iii) the French press was set to 16,000 psi for making IMVs; and (iv) the three step sucrose gradient used for IMV isolation was 2.1 M, 1.5 M, and 0.5 M.

**Spheroplasts for imaging.** Spheroplasts were prepared using the same DADE strains transformed with pTatAB^mNeonC or pTatABC^mNeon that were used to make IMVs. As described earlier[7] and indicated in the previous section, spheroplasts were made before cell rupture by French press to make IMVs. To image such spheroplasts, a starter culture of modified low-salt LB media (5 mL) was grown overnight at 37 °C, and then was used to inoculate a fresh 5 mL culture in the morning. The new culture was incubated at 37 °C until $OD_{600} \approx 0.6$, whereupon protein overproduction was initiated with 0.7% arabinose, and the induced culture was incubated for 1 h at 37 °C. Spheroplasts were made from the recovered cells ($5000 \times g$, 10 min, 4 °C). The spheroplast mixture was diluted ~1:5000 with the buffer used to make spheroplasts[7], thus diluting out the EDTA and lysozyme. The diluted spheroplasts were added to a coverslip flow chamber and imaged under the microscope.

**In vitro transport assays.** Transport of pre-SufI(IAC)^Alexa647 into IMVs was assayed essentially as described previously[7,17,45]. In short, transport reactions (35 µL) included IMVs ($OD_{280} = 5.0$) and pre-SufI(IAC)^Alexa647 (50 nM) in Transport Buffer (TB; 25 mM MOPS, 25 mM MES, 5 mM MgCl$_2$, 50 mM KCl, 200 mM sucrose, pH 8.0) with 1 µg/µL of BSA. Transport of substrate into the IMVs was initiated by the addition of 4 mM NADH, and continued for 20 min at 37 °C. Untransported (external) substrate was digested by incubating with 0.57 µg/µL proteinase K for 30 min at RT. Digestions were quenched by the addition of PMSF (5 mM) and SDS Sample Buffer (100 mM Tris-HCl, 4.5 M Urea, 2% SDS, 5% glycerol, 0.02% bromophenol blue, 0.2% βME, pH6.8). Samples were heated at 95 °C for 10 min and then analyzed on an 12% SDS-PAGE gel.

**Western blotting.** Western blots were performed using PVDF membranes, with a transfer time of 110 min at 400 mA. The transfer tank was chilled by placing it in a bucket of ice during the transfer process. TatB and TatC were detected using anti-TatB and anti-TatC primary antibodies[67] (1:2500 dilution), and polyclonal goat anti-rabbit IgG-HRP secondary antibodies (1:10,000 dilution; Santa Cruz Biotech, Inc).

**Protein analytical methods.** Concentrations of purified proteins were determined by the densitometry of bands on Coomassie Blue R-250 stained SDS-PAGE gels using carbonic anhydrase as a standard and a ChemiDoc MP imaging system (Bio-Rad Laboratories). Fluorescent proteins were detected by direct in-gel fluorescence imaging using the same ChemiDoc imaging system. Alexa647 and mNeon concentrations were determined using $\varepsilon_{650} = 270,000$ cm$^{-1}$ M$^{-1}$ and $\varepsilon_{506} = 116,000$ cm$^{-1}$ M$^{-1}$[42], respectively. Western blot bands were visualized by chemiluminescence using the Clarity Max Western blotting kit (Bio-Rad Laboratories, #1705062) and the ChemiDoc imaging system. IMV concentrations were determined as the $A_{280}$ in 2% SDS[7].

**Dynamic light scattering.** DLS experiments were performed on Zetasizer Nano (Malvern Panalytical). IMV samples were diluted 1:100 with 50 mM Tris, 150 mM NaCl, pH 7.5 for analysis.

**Coverslip preparation.** Coverslips ($24$ mm $\times 60$ mm; VWR, #16004-312) were cleaned by sonicating in distilled water for 10 min, followed by sonicating in

acetone for 10 min. After air-drying in a hood, the coverslips were plasma cleaned in argon for 5 min on the high setting (Harrick Plasma, Model PDC-001). The plasma-cleaned slides were mounted in home-machined aluminum microscope holders and held in place by high-vacuum grease (Dow Corning, VWR #59344-055). Flow chambers were constructed from strips of double-sided tape and a top coverslip (10.5 mm × 22 mm; Electron Microscopy Sciences, #72191-22).

**Microscopy**. Single particle photobleaching analyses were performed with a Nikon Eclipse Ti using a ×100 oil-immersion objective (Nikon Apo TIRF, 1.49 NA) and narrow-field epifluorescence excitation[47] with 488 and 640 nm diode lasers (Coherent). Images were magnified (×1.5 tube lens) and recorded using an EMCCD (Evolve Delta; Photometrics) or CMOS (Prime 95B; Photometrics) camera. The focus was adjusted, focus-lock engaged, and the coverslip translated such that the recordings for the samples analyzed captured the photobleaching profile from the moment samples were first illuminated. This insured that no photobleaching events were missed. Exposures were 100 ms with a recording length of 500–600 frames.

All buffers that were used in the single particle imaging protocol were passed through a 0.22 μm syringe filter, which substantially reduced the number of fluorescent background particles. The purified mNeon standards were diluted with Dilution Buffer (50 mM Tris-HCl, 150 mM NaCl, pH 7.5), incubated in flow chambers for a minimum of 5 min, washed with two-lane volumes of Dilution Buffer (~20 μL), and then imaged. For CellMask staining, IMVs (OD$_{600}$ = 5.0) containing TatB$^{mNeon}$ and TatC$^{mNeon}$ were first incubated with 0.1× CellMask Deep Red Plasma Membrane Stain (ThermoFisher Scientific, # C10046) for 15 min at RT. The IMV solutions were then diluted with Dilution Buffer, incubated in a flow chamber for a minimum of 5 min, washed with two lane volumes of Dilution Buffer, and imaged. The same protocol was followed for staining IMV preparations with 100 pM BODIPY FL C16 (ThermoFisher Scientific, #D3821) and 2× FluoVolt (ThermoFisher Scientific, #F10488).

**Image analysis**. Fluorescent spots in photobleaching videos were selected manually using ROI Manager in the Fiji package of ImageJ[68]. For the mNeon standards, all spots were selected. For IMVs, only spots that colocalized with CellMask were selected (spots visualized in both the 488 and 640 nm excitation channels). For every 10 × 10 pixel ROI chosen, an adjacent area of the same size with no fluorescence was chosen for background subtraction. Intensities were obtained via the multi-measure function of ROI Manager. For visual step counting, photobleaching profiles were plotted using MATLAB.

For PDD analysis[48], the intensity values from photobleaching traces were used to generate pairwise difference curves using a custom MATLAB script. The pairwise differences for each trace were then binned, normalized, and plotted as a frequency histogram. The position of the last peak (highest photon level) from the histogram was used to estimate the number of photobleaching steps in the corresponding trace, as described in Supplementary Fig. 9b.

**Simulations**. Step expectation values were calculated assuming that Tat receptor complexes partitioned into IMVs following a Poisson distribution and the active fluorophores within receptor complexes and mNeon dimer and tetramer standards followed a binomial distribution, where the FDE defines the probability that the fluorophore is actively fluorescent (Supplementary Software Files 1 and 2). Standard deviations were obtained from 5000 independent trials of $N$ measurements (where $N$ = number of experimentally visualized spots analyzed) using Microsoft Excel with the RiskAMP Monte Carlo plugin (Structured Data LLC) (Supplementary Software File 3). Photon frequency histograms for various numbers of photobleaching steps (Supplementary Fig. 9b) were estimated with Supplementary Software File 4.

**Statistics and reproducibility**. The main conclusions of this paper summarized in Fig. 8 and Supplementary Table 1 were obtained from three data sets each for TatB$^{mNeon}$ and TatC$^{mNeon}$. As discussed and indicated within the figures, the data were fit to various oligomerization models, and the best model was determined by the lowest RMSD.

**Reporting summary**. Further information on research design is available in the Nature Research Reporting Summary linked to this article.

## Data availability

Plasmids pET-28a_mScarlet, pH6-mNeon, pH6-mNeonC, pH6-2xmNeon, pH6-2xmNeonC, pBAD_TatAB$^{mCherry}$C, pTatAB$^{mNeon}$C, pTatA$_k$B$_x$C$_s$, and pTatABC$^{mNeon}$ are available from Addgene (IDs# 178460-178465, 178543-45). The authors declare that all other data supporting the findings of this study are available within the paper and its supplementary information files. The source data behind graphical figures is available in Supplementary Data 1. Uncropped and unedited gel and blot images are available in

Supplementary Figs. 15–17. Materials are available from the corresponding author upon reasonable request.

## Code availability

Supplementary Software Files 1–4, as described in the Methods, are available online.

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

## Acknowledgements

We thank T.L. Yahr for antibodies to TatB and TatC, T. Palmer for the DADE strain, U. Bageshwar for constructing the arabinose-resistant DADE strain, S. Hamsanathan for assistance with preliminary studies, and B. Pettijohn for making the parent pTatA$_k$B$_x$C$_s$ plasmid. This research was supported by the National Institutes of Health (GM116995 to SMM).

## Author contributions

SMM conceived the approach, AS purified materials, collected and analyzed data, and wrote the manuscript, RC helped with microscope experiments and data analysis, SMM developed the simulations, provided advice, analyzed data, and edited the manuscript.

## Competing interests

The authors declare no competing interests.
