## [Peer Review File · Communications Biology]

Reviewers' comments:

Reviewer #1 (Remarks to the Author):

In their manuscript, Sharma et al. report photobleaching-based counts of mNeonGreen (mNeon) numbers in inverted membrane vesicles (IMVs) containing TatABC systems with fusions of mNeon to either TatB or TatC. After fitting Poisson distributions to the experimental data, and after including a variable to take non-fluorescent tags into account, they conclude that functional TatBC receptor complexes consist of four TatBC units (i.e. eight subunits), which they term "tetramers".

The question is important for the Tat field and the approach is novel. The study reports an elaborate quantification of fluorescent probes in IMVs. The quality of the text is high and the experiments were technically challenging. However, I see a problem with the use of the Poisson distribution for the fits, and with the inclusion of a variable factor that accounts for a quite high portion of non-fluorescent mNeonGreen tags in these fits. The interpretation of the data is also hampered by the fact that the authors use a very strong expression system. Using their established expression system for in vitro studies, they shortened the growth under inducing conditions from 4 to 1 h to achieve an abundance of Tat components that was suitable for their analyses, instead of using an expression system designed for steady state low levels of tagged Tat systems. Another drawback is the observation of a significant population of fluorophores that are not associated with membranes, although the tagged components should be stably membrane-integrated proteins. Taken these aspects into account, important issues need to be clarified to improve the manuscript.

Here are the main aspects in more detail:

(1) The pBAD-based expression system with 0.7 mM arabinose as inducer can result in very strong over-production of the target proteins. I am not sure, whether the DADE strain, which is an *araD* mutant, was ever selected for arabinose resistance by the group (there is no mentioning of this). The shortening of the induction time to 1 hour served to achieve fluorescent protein abundances that are low enough for the analyses, but a large number of membrane proteins will be present that is not yet incorporated in functional Tat complexes. The pBAD system is problematic in that strain. It would have been better to generate a constitutive lower level expression system for these analyses.

(2) The Poisson distribution can be applied if the Tat complexes are randomly distributed in the cytoplasmic membrane. Such a distribution cannot be assumed: TatC has been detected only in few positions per cell (Berthelmann et al. JBC 2008, Rose et al. PLOS One 2013, Alcock et al. PNAS 2013). If a cell generates in average 200 IMVs (page 13), and if active Tat receptor complexes would be in 18-33 % of these IMVs (page 8), then there would exist at least roughly 40-60 independent receptor-positions per cell (that might contain one or more receptors). It therefore can be that the authors also counted associations other than functional TatBC complexes.

(3) If I understood it right, the best fits suggest FDEs in the range of 0.5 (Fig.7d). Aren't these too low values? Are there really that many mNeonGreen fusions non-fluorescent? Why is the FDE not more or less the same as for controls (ca.0.8, Fig.6)? What justifies to use the FDE as a parameter that can be fitted, and not as a constant that can be derived from the control experiments? The authors surely can explain this better.

(4) The best fit predicts 40-50% IMVs to be empty in case of tetramers (Fig. 7e). However, experimentally observed were 67-82%. Is that o.k.? Or isn't the outcome an argument against the use of the Poisson distribution in combination with the FDE parameter for fitting? The discussion on page 12 mentions that a non-homogeneous distribution may be the reason. The authors recognized this possible explanation for the inconsistent data already on page 9. However, if the non-homogeneous distribution caused problems with the Poisson fit (in addition to the FDE), the postulation of the tetramer of TatBC dimers might be wrong. This should be carefully considered.

(5) The authors recognized and mentioned (page 7) that the fluorescence resolution does not suffice to address whether the fluorescence originates from interacting proteins. They therefore should draw their conclusions more carefully, as they have no direct evidence for an interaction.

(6) Large fluorescent associations in IMVs were regarded as "aggregates" and not included in the analyses (page 8). It may well be that these were the few active, usually polar associations per cell that have been seen in several previous studies. If so, then the authors focused on the wrong signals. It would be good, if the authors could consider this and have a look at these larger assemblies. Maybe they can provide evidence for or against their functionality.

(7) There must be some explanation for the many fluorescent spots in regions without membrane vesicles. The manuscript does not really give a reasonable explanation. The vesicles were purified by a step gradient and there should not be membrane-free proteins (ca. 1.3 g/ml) in an IMV preparation. Non-tagged TatB and TatC are known not to be released from membranes by cell disruption or even carbonate washes. What explains these spots? The inclusion of "membrane-free" Tat systems in Fig. 1 is unrealistic. I am puzzled. Were the tagged components not stably membrane-integrated? This should be analyzed by carbonate washes. If many fluorescent proteins are not correctly membrane integrated under these expression conditions, then the study analyzed an artefact. Considering the large portion of mNeonGreen-tagged Tat proteins not associated with lipid vesicles (35-70%, page 8), this may well be the case. Alternatively, but less likely, it might be that the tag was cleaved off by proteases after IMV preparation. Were these fluorescent spots therefore single mNeonGreen proteins, as revealed by photobleaching, or did they behave like the membrane-associated spots?

Minor aspects:

Could the authors discuss the differences in the TatC-mNeon signal pattern in Suppl. Fig 2B and 2C? Could the authors give the reason for the strange "YTRVP"-linker in TatB(mNeon)?

Page 3: The consensus motif cannot end with an X and includes at least one residue preceding the RR. Published are either the initial pattern S/T-R-R-X-F-L-K or the generalized pattern Z-R-R-X-Phi-Phi, in which Z and Phi stand for polar and hydrophobic residues, respectively.

Page 5: The authors state that the large Tat complexes obtained upon detergent extraction can over-estimate oligomer size mainly due to "co-purifying molecules, such as lipids". Most of these analyses were BN-PAGE analyses, and I would therefore not say "co-purifying molecules, such as lipids" but rather "lipid and detergent content of the analyzed micelles".

Page 9, line 13: Most likely "Supplementary Fig 2." should be "Supplementary Fig. 3".

In context of the discussion at the top of page 12, could the authors comment on the fluorophore formation kinetics of mNeonGreen (I guess it is known), which might partially account for variable amounts of non-fluorescent mNeonGreen in their analyzed samples?

Throughout: The term "tetramer" is misleading for a heterooligomeric complex consisting of eight subunits. This is especially confusing in Figures that refer to oligomer size (Fig.7c,d,e).

The authors could consider that large tags such as mNeonGreen may influence the interactions of Tat system components and the assembly of Tat receptor complexes, which may have affected the outcome of this study. Certainly others used the same fusion technique, but this does not mean that it has no effect.

In conclusion, this manuscript shows very interesting data, but the calculations may have been misleading, and the nature of the fusion proteins of the analyzed vesicles is unclear. Some more controls should be added to address the second aspect (carbonate washes, analysis of the larger associations, analyses of spots outside the IMVs) and alternative scenarios and explanations should be included for the first aspect. In the end, the authors have to carefully reassess their data and they may come to alternative conclusions in the course of a revision.

Reviewer #2 (Remarks to the Author):

The Tat protein transport system is one of the most widely distributed and important protein transport machines in biology. Nevertheless, it has proved very difficult to elucidate structural information about the system. Two proteins TatB and TatC make up a substrate receptor/core complex onto which another TatA component assembles. Even 20 years after the TatBC complex was first isolated the subunit stoichiometry has still not been established due to technical challenges. This paper reports a study in which the TatBC complex stoichiometry is assessed in native membranes by step photobleaching of fluorescent protein fusions of the subunits. The authors find a 4 TatB : 4 TatC composition for the TatBC complex. Determining this number is a significant advance in the field and provides constraints on mechanistic models for Tat transport.

This is experimentally challenging work. The study is carefully carried out and the paper is well presented. This paper would be suitable for publication in *Communications Biology* if the authors can address the points below.

Major Points

[1] The raw photobleaching step data in Fig.5 suggests peaks in the frequency distribution at multiples of three. However, vesicles with step sizes that are not divisible by three are also present. The authors assume that these arise from a failure to mature all fluorophores present, resulting in a stochastically distributed undercounting of subunits. The distributions are then modelled taking this phenomenon into account resulting in a best fit to the data for tetramer complexes. This correction is entirely reasonable based on the known incomplete maturation yields of fluorescent proteins. However, there is another potential origin of the 'incorrect' step measurements and that is data noise caused by issues in correctly assigning steps. This would broaden the true stoichiometry peaks in the distribution. I think this is likely to be an issue because an examination of the raw photobleaching traces in Fig. 3b (which presumably have not been picked because they are the worst looking data) shows that the assignment of stoichiometry is quite subjective. You could easily call the 'tetramer' as a trimer, the 'hexamer' as a pentamer, and the 'octamer' as a heptamer, for instance. Can the authors exclude this possibility that the deviations from an ideal distribution are predominantly due to step assignment error? At the very least this does need to be addressed in the text.

[2] The authors report that a large proportion (35-70%) of the protein fluorescence is not associated with lipid vesicles. This is odd and concerning. As the authors point out this behaviour would not be expected of an integral membrane protein like TatC or TatB, but the authors do not provide an alternative explanation. Can they provide a plausible explanation? Looking at the raw images shown in Fig. 3a the membrane staining gives the impression of a biased distribution in the field of view which raises the possibility that the laser illumination is not constant across the image and thus may be missing some vesicles. Is this possible?

Other points

Pg 6. 'topologically inverted' rather than 'typologically inverted'.

Pg12. The authors state that the TatBC oligomeric state cannot be an odd number because TatC-TatC dimer fusions are functional. In my opinion that evidence does not definitively exclude an odd stoichiometry. For example, for the lowest odd oligomer – a trimer – you could have a situation where one dimer occupies two positions in the trimer and the second dimer has one TatC copy in the trimer complex and one TatC copy dangling on the outside of the complex, or there could be three TatC dimers with only one TatC copy in each dimer forming the functional complex. The authors should soften their certainty here and just say that those observations are most consistent with even stoichiometry rather than saying that they completely exclude an odd stoichiometry.

Pg 13. The authors guess that only a small proportion of the IMVs containing TatBC that they visualise

are competent for Tat transport. Since their transport substrate is labelled with a far-red dye could they not test this hypothesis by imaging the vesicles and counting the proportion that have taken up the substrate protein?

Fig. 3b. I think it is misleading to label the graphs 'tetramer', 'hexamer', and 'octamer' because the traces do not necessarily arise from a single oligomer. Better '4-steps', '6-steps', '8-steps'.

Fig 7 and SI Fig 7. The Poisson Mean quoted in the figures is presumably the mean of the distribution of complexes in the model (assuming the particular oligomeric state tested) and not the actually mean of the step distribution shown in the Figure. This is going to confuse some readers so I would recommend that this is explicitly explained. I also think the authors might be more explicit that their criterion for picking the best model is that they are picking the model with the lowest residuals between the fit and that data.

Re: Manuscript COMMSBIO-21-3190-T

"Oligomerization state of the functional bacterial twin arginine translocation (Tat) receptor complex"

A point-by-point response to the Reviewer's comments is provided below. Comments from the Reviewers are in *italics*, and our responses follow in **bold**. Page and line numbers in our responses below refer to the 'tracked changes' version of the revised manuscript. An additional 'clean' version of the manuscript is also provided.

REVIEWER 1:

MAJOR POINTS:

1. The pBAD-based expression system with 0.7 mM arabinose as inducer can result in very strong over-production of the target proteins. I am not sure, whether the DADE strain, which is an araD mutant, was ever selected for arabinose resistance by the group (there is no mentioning of this). The shortening of the induction time to 1 hour served to achieve fluorescent protein abundances that are low enough for the analyses, but a large number of membrane proteins will be present that is not yet incorporated in functional Tat complexes. The pBAD system is problematic in that strain. It would have been better to generate a constitutive lower level expression system for these analyses.

We thank the Reviewer for pointing out the need for selecting an arabinose resistance strain when using an arabinose expression system. Our lab has indeed selected an arabinose resistance strain of DADE, which was used for all of our arabinose induction experiments. This is now noted in the Methods section (p. 19, lines 382-384).

2. The Poisson distribution can be applied if the Tat complexes are randomly distributed in the cytoplasmic membrane. Such a distribution cannot be assumed: TatC has been detected only in few positions per cell (Berthelmann et al. JBC 2008, Rose et al. PLOS One 2013, Alcock et al. PNAS 2013). If a cell generates in average 200 IMVs (page 13), and if active Tat receptor complexes would be in 18-33 % of these IMVs (page 8), then there would exist at least roughly 40-60 independent receptor-positions per cell (that might contain one or more receptors). It therefore can be that the authors also counted associations other than functional TatBC complexes.

We thank the Reviewer for these thoughtful comments. During our IMV preparation, the bacterial cells were first treated with lysozyme and EDTA to remove the cell wall and generate spheroplasts. IMVs were made by French pressure treatment of these spheroplasts. Thus, it is the distribution of Tat complexes within spheroplasts, not within intact cells, that is crucial for the Poisson distribution assumption. Images of these spheroplasts suggest a fairly homogenous distribution of Tat proteins in the membrane (Supplementary Fig. 14), supporting our random distribution assumption and that the Tat complexes are Poisson distributed into the IMVs (now discussed on p. 16, lines 337-342).

3. If I understood it right, the best fits suggest FDEs in the range of 0.5 (Fig.7d). Aren't these too low values? Are there really that many mNeonGreen fusions non-fluorescent? Why is the FDE not more or less the same as for controls (ca.0.8, Fig.6)? What justifies to use the FDE as a parameter that can be

fitted, and not as a constant that can be derived from the control experiments? The authors surely can explain this better.

We thank the Reviewer for pointing out the need for clarity regarding the FDE. It was, in fact, our original assumption that the FDE would be constant, and that we could therefore use multiple control constructs to determine the FDE, and then use this value to fit the photobleaching data for Tat fusion proteins. However, this assumption was simply not valid, as even the mNeon dimer and tetramer data could not be fit with a single FDE. To illustrate this point, we now compare the FDE obtained from photobleaching data for genetic mNeon dimers, and for an mNeon disulfide dimer generated from two monomeric mNeon proteins. These dimer constructs have similar molecular weights, except that one was expressed and matured as a polypeptide twice the size of the other. However, their FDEs were different (0.77 vs. 0.82). Moreover, the genetic dimer had the same FDE as the mNeon tetramer, which was formed as a disulfide dimer of two genetic mNeon dimers (p, 12, lines 244-254). Thus, the variable FDE for different fusion proteins is reasonable. In short, it is essential to consider the FDE as a fit parameter, and not as a constant. We have compared all the FDEs obtained (Supplementary Fig. 13) and further discussed the issue of FDE variability (p. 13, lines 270-277).

4. The best fit predicts 40-50% IMVs to be empty in case of tetramers (Fig. 7e). However, experimentally observed were 67-82%. Is that o.k.? Or isn't the outcome an argument against the use of the Poisson distribution in combination with the FDE parameter for fitting? The discussion on page 12 mentions that a non-homogeneous distribution may be the reason. The authors recognized this possible explanation for the inconsistent data already on page 9. However, if the non-homogeneous distribution caused problems with the Poisson fit (in addition to the FDE), the postulation of the tetramer of TatBC dimers might be wrong. This should be carefully considered.

We thank the Reviewer for these comments and for asking us to reassess the high number of empty IMVs. A key assumption here is that the hydrophobic CellMask dye identifies IMVs, and only IMVs. To examine this assumption, we increased our analyzed dataset size and compared the localization of CellMask with two other hydrophobic dyes, which were also assumed to partition into the IMV membrane bilayer. Surprisingly, ~40-60% of the CellMask dye localizations did not colocalize with the other dyes in pairwise comparisons (Fig. 3c). Therefore, we conclude that the number of CellMask fluorescent spots likely represents an overestimate of the number of IMVs. Nonetheless, use of the CellMask dye still provides a substantial benefit in that it allows for an initial filtering of the data – by only analyzing CellMask colocalized spots, the small number of contaminating fluorescent spots and soluble fluorescent proteins are effectively eliminated/reduced. As the two tested dyes (BODIPY-FL C16 and FluoVolt) both require 488 nm excitation, we were unable to colocalize these dyes with the mNeonGreen labeled Tat proteins. In short, the number of empty IMVs estimated from the best fit to the photobleaching data is considered a better estimate than the value determined from the CellMask data. Two paragraphs are dedicated to these data and conclusions (pp. 8-10, lines 157-195).

The homogeneity assumption was addressed in our response to major point 2.

5. The authors recognized and mentioned (page 7) that the fluorescence resolution does not suffice to address whether the fluorescence originates from interacting proteins. They therefore should draw their conclusions more carefully, as they have no direct evidence for an interaction.

Our statement on fluorescence resolution was intended to clarify our approach, namely, that our strategy was not to spatially resolve separate complexes, e.g., which can and has been done in intact cells. IMVs are simply too small for such an approach. Instead, we fit our data to distinct oligomerization models – strong interactions are requisite for and distinguish these models, which yield experimentally distinguishable characteristics. We clarify our approach and assumptions on p. 8, lines 153-156. The additional description of our approach in the Experimental Design section further expands on our experimental strategy (p. 6, lines 104-115).

6. Large fluorescent associations in IMVs were regarded as “aggregates” and not included in the analyses (page 8). It may well be that these were the few active, usually polar associations per cell that have been seen in several previous studies. If so, then the authors focused on the wrong signals. It would be good, if the authors could consider this and have a look at these larger assemblies. Maybe they can provide evidence for or against their functionality.

The ‘active’ form of Tat receptor complex is an important issue and a focus of our current work, which will be reported in a subsequent paper. However, what we can report at this time is that some of the IMVs with low mNeon fluorescent intensities do indeed transport precursor proteins. Thus, it does not appear that Tat ‘aggregates’ are essential for transport.

7. There must be some explanation for the many fluorescent spots in regions without membrane vesicles. The manuscript does not really give a reasonable explanation. The vesicles were purified by a step gradient and there should not be membrane-free proteins (ca. 1.3 g/ml) in an IMV preparation. Non-tagged TatB and TatC are known not to be released from membranes by cell disruption or even carbonate washes. What explains these spots? The inclusion of “membrane-free” Tat systems in Fig. 1 is unrealistic. I am puzzled. Were the tagged components not stably membrane-integrated? This should be analyzed by carbonate washes. If many fluorescent proteins are not correctly membrane integrated under these expression conditions, then the study analyzed an artefact. Considering the large portion of mNeonGreen-tagged Tat proteins not associated with lipid vesicles (35-70%, page 8), this may well be the case. Alternatively, but less likely, it might be that the tag was cleaved off by proteases after IMV preparation. Were these fluorescent spots therefore single mNeonGreen proteins, as revealed by photobleaching, or did they behave like the membrane-associated spots?

We thank the Reviewer for raising these concerns. The ‘membrane-free’ mNeon fluorescence does not arise from cleaved mNeon-Tat fusions, as shown by Western blots (Supplementary Fig. 2), which are discussed on p. 7 (lines 141-143). To address whether the membrane-free mNeon-Tat proteins arise from protein that had not yet been incorporated into the membrane, we spun down cells to remove the arabinose inducer and then continued growth for 2 hours without arabinose, providing time for already produced protein to incorporate into the membrane (‘2 h chase’). The mNeon-Tat fusions from such cells yielded a substantially increased colocalization with CellMask (Fig. 3b), suggesting that the mNeon-Tat fusions that do not colocalize with CellMask had not yet been incorporated into the membrane. Although in principle the IMVs should be separable from soluble protein in the sucrose gradient purification step, the IMVs were recovered from near the top step of the sucrose gradient, and thus (apparently) poorly resolved from the soluble proteins. By analyzing only those complexes that colocalized with CellMask, we assume that we have eliminated most, if not all, of the fluorescent Tat proteins that had not yet integrated into the membrane. These issues are now discussed (pp. 8-9, lines 157-180).

MINOR POINTS:

1. *Could the authors discuss the differences in the TatC-mNeon signal pattern in Suppl. Fig 2B and 2C?*

TatC is highly hydrophobic, which presents challenges in quantitative transfer in Western blots. However, we have improved our protocol by increasing the transfer time and by cooling the blotting chamber during transfer (p. 23, lines 489-491). We now include a new Western blot image of TatC^{mNeon} (Supplementary Fig. 2B), which confirms that there is no cleaved mNeon in the sample. The slight difference in apparent molecular weight for TatCmNeon is a result of different gel running conditions, namely, 4.5 M urea was added to the samples in Supplementary Fig. 2a and Supplementary Fig. 2b, but not for those in Supplementary Fig. 2c (now noted in the figure caption, p. 46, lines 1024-1026).

2. *Could the authors give the reason for the strange "YTRVP"-linker in TatB(mNeon)?*

The RVP amino acid sequence reflects an introduced KpnI restriction site. This is now indicated in the Supplementary Fig. 1 caption (p. 45, lines 1011-1012).

3. *Page 3: The consensus motif cannot end with an X and includes at least one residue preceding the RR. Published are either the initial pattern S/T-R-R-X-F-L-K or the generalized pattern Z-R-R-X-Phi-Phi, in which Z and Phi stand for polar and hydrophobic residues, respectively.*

We thank the Reviewer for pointing this out. We now indicate that the consensus motif is (S/T)RRXFLK (p. 3, line 44).

4. *Page 5: The authors state that the large Tat complexes obtained upon detergent extraction can overestimate oligomer size mainly due to "co-purifying molecules, such as lipids". Most of these analyses were BN-PAGE analyses, and I would therefore not say "co-purifying molecules, such as lipids" but rather "lipid and detergent content of the analyzed micelles".*

Changed as requested (p. 5, line 85).

5. *Page 9, line 13: Most likely "Supplementary Fig 2." should be "Supplementary Fig. 3".*

Yes, the Reviewer is correct. However, due to Supplementary Figure renumbering, the figure is now Supplementary Fig. 8 (p. 11, line 236).

6. *In context of the discussion at the top of page 12, could the authors comment on the fluorophore formation kinetics of mNeonGreen (I guess it is known), which might partially account for variable amounts of non-fluorescent mNeonGreen in their analyzed samples?*

For fluorescent proteins to become fluorescent, they must both fold properly and oxidize to form the chromophore, which is a cyclization reaction involving multiple amino acids. It is not

known which step is most influenced in the different mNeonGreen constructs – this is beyond the scope of this study. However, as discussed in our response to major point 3, we have expanded our discussion regarding the variable FDE at two locations within the text.

7. Throughout: The term "tetramer" is misleading for a heterooligomeric complex consisting of eight subunits. This is especially confusing in Figures that refer to oligomer size (Fig.7c,d,e).

We understand the confusion; however, trimer, tetramer, etc. is the common terminology within the field. The receptor complex is generally agreed to be comprised of an oligomer of TatBC heterodimers; nonetheless, there is likely some TatA (and/or TatE) within the receptor complex as well, further complicating the issue. This is described extensively in the second paragraph of the introduction (pp. 3-4, lines 51-67). To increase clarity, we now indicate that trimer, tetramer, etc. refers to the oligomeric state of TatBC heterodimers within the Tat receptor complex (p. 4, lines 65-67).

8. The authors could consider that large tags such as mNeonGreen may influence the interactions of Tat system components and the assembly of Tat receptor complexes, which may have affected the outcome of this study. Certainly others used the same fusion technique, but this does not mean that it has no effect.

We thank the Reviewer for bringing up this point. We agree that any external tag could negatively influence biological function, especially for a tag of significant size within an oligomeric assembly. However, transport using TatB^{mNeon} was similar to wildtype TatB (Figs. 2a & 2b), TatB^{mNeon} and TatC^{mNeon} yielded similar transport under the same conditions (Fig. 2a), and mCherry fusions with TatB and TatC also yielded similar transport efficiencies (Whitaker, et al 2012 *J. Biol. Chem.* 287:11252). While such experiments do not indicate that all aspects of the transport cycle are unaffected, we think it reasonable to assume that the oligomerization state of the receptor complex is not perturbed. This is now discussed on pp. 7-8, lines 143-147.

REVIEWER 2:

MAJOR POINTS:

1. The raw photobleaching step data in Fig.5 suggests peaks in the frequency distribution at multiples of three. However, vesicles with step sizes that are not divisible by three are also present. The authors assume that these arise from a failure to mature all fluorophores present, resulting in a stochastically distributed undercounting of subunits. The distributions are then modelled taking this phenomenon into account resulting in a best fit to the data for tetramer complexes. This correction is entirely reasonable based on the known incomplete maturation yields of fluorescent proteins. However, there is another potential origin of the 'incorrect' step measurements and that is data noise caused by issues in correctly assigning steps. This would broaden the true stoichiometry peaks in the distribution. I think this is likely to be an issue because an examination of the raw photobleaching traces in Fig. 3b (which presumably have not been picked because they are the worst looking data) shows that the assignment of stoichiometry is quite subjective. You could easily call the 'tetramer' as a trimer, the 'hexamer' as a pentamer, and the 'octamer' as a heptamer, for instance. Can the authors exclude this possibility that the deviations from an ideal distribution are predominantly due to step assignment error? At the very least this does need to be addressed in the text.

We thank the Reviewer for the careful and balanced assessment of our work, and for the recognition of the many challenges faced. To eliminate (or substantially reduce) the subjectivity in assigning steps to photobleaching traces, we have attempted to analyze our data using multiple algorithms. In general, we found that existing algorithms tend to be specific for the characteristics of the experiments or fluorescent proteins for which they are designed. For example, the 'change points' function in MATLAB was unable to accommodate blinks during step finding, and the quickPBSA approach (Hummert, et al., 2021, *Mol. Biol. Cell*, 32:ar35) was not able to satisfactorily model mNeonGreen's blinking rate, leading to underfitting or overfitting of the photobleaching profile. However, we found that the pairwise difference distribution (PDD) function was readily implemented. Notably, PDD analysis yielded the same conclusions as our original analysis (Fig. 8, Supplementary Fig. 12). Use of this PDD analysis is now mentioned on p. 10 (lines 212-214) and described in the Methods (p. 26, lines 543-548).

2. The authors report that a large proportion (35-70%) of the protein fluorescence is not associated with lipid vesicles. This is odd and concerning. As the authors point out this behaviour would not be expected of an integral membrane protein like TatC or TatB, but the authors do not provide an alternative explanation. Can they provide a plausible explanation? Looking at the raw images shown in Fig. 3a the membrane staining gives the impression of a biased distribution in the field of view which raises the possibility that the laser illumination is not constant across the image and thus may be missing some vesicles. Is this possible?

We understand the Reviewer's concerns. The colocalization issue and the over-estimation of IMV counts were addressed in our responses to major points 4 and 7 of Reviewer 1.

We have used narrow-field illumination (rather than wide-field; p. 8, line 163) to enhance the signal-to-noise ratio for single molecule imaging. Thus, only the central area of the microscope field was illuminated. This is now clarified in Fig. 3a itself, and in the caption.

MINOR POINTS:

1. Pg 6. 'topologically inverted' rather than 'typologically inverted'.

This has been corrected (p. 6, line 110).

2. Pg12. The authors state that the TatBC oligomeric state cannot be an odd number because TatC-TatC dimer fusions are functional. In my opinion that evidence does not definitively exclude an odd stoichiometry. For example, for the lowest odd oligomer – a trimer – you could have a situation where one dimer occupies two positions in the trimer and the second dimer has one TatC copy in the trimer complex and one TatC copy dangling on the outside of the complex, or there could be three TatC dimers with only one TatC copy in each dimer forming the functional complex. The authors should soften their certainty here and just say that those observations are most consistent with even stoichiometry rather than saying that they completely exclude an odd stoichiometry.

We thank the Reviewer for suggesting this important alternate interpretation of the TatC dimer fusion data. As suggested, we have softened our conclusions, indicating that an even stoichiometry is more likely (p. 15, lines 314-317).

3. Pg 13. *The authors guess that only a small proportion of the IMVs containing TatBC that they visualise are competent for Tat transport. Since their transport substrate is labelled with a far-red dye could they not test this hypothesis by imaging the vesicles and counting the proportion that have taken up the substrate protein?*

We thank the Reviewer for this suggestion. As we indicated in our response to Reviewer #1 (major point 6), the transport efficiency of individual IMVs is a focus of our current work and will be the topic of a subsequent paper. There are multiple complications to a proper analysis, including the need to firmly attach the IMVs to the coverslip surface so they do not wash away, and verifying that external substrate (i.e., bound to the Tat receptor complex and the membrane surface) is washed away. However, preliminary indications are that the number of active IMVs seems to be < 20%.

4. *Fig. 3b. I think it is misleading to label the graphs 'tetramer', 'hexamer', and 'octamer' because the traces do not necessarily arise from a single oligomer. Better '4-steps', '6-steps', '8-steps'.*

We agree and thank the Reviewer for this excellent suggestion. Figures 4 and 5 now indicate the number of steps rather than the oligomer size.

5. *Fig 7 and SI Fig 7. The Poisson Mean quoted in the figures is presumably the mean of the distribution of complexes in the model (assuming the particular oligomeric state tested) and not the actually mean of the step distribution shown in the Figure. This is going to confuse some readers so I would recommend that this is explicitly explained. I also think the authors might be more explicit that their criterion for picking the best model is that they are picking the model with the lowest residuals between the fit and that data.*

We agree with both the meaning of Poisson mean and the criteria for choosing the best fit. The text was updated (p. 13, lines 261-263 and p. 14, lines 302-305) to clarify these points.

REVIEWERS' COMMENTS:

Reviewer #1 (Remarks to the Author):

The very detailed answers were convincing and the authors did a lot of efforts to improve the manuscript. I therefore fully support the publication of this study.

I only found some typing errors. The authors forgot to delete the words "aggregation of" on page 5, line 84/85, and they have a spelling error in "Octamer Fit" in Supplemental Figure 10f, and "octamer fits" in the footnote for the Supplementary Table 1. There may be also one "mNeon" too much the legend of Fig.7b.

Reviewer #2 (Remarks to the Author):

The authors have addressed the concerns I set out in my original review.